# TOWARDS MULTI-SENSE CROSS-LINGUAL ALIGNMENT OF CONTEXTUAL EMBEDDINGS

## ABSTRACT

Cross-lingual word embeddings (CLWE) have been proven useful in many cross-lingual tasks. However, most existing approaches to learn CLWE including the ones with contextual embeddings are sense agnostic. In this work, we propose a novel framework to align contextual embeddings at the sense level by leveraging cross-lingual signal from bilingual dictionaries only. We operationalize our framework by first proposing a novel sense-aware cross entropy loss to model word senses explicitly. The monolingual ELMo and BERT models pretrained with our sense-aware cross entropy loss demonstrate significant performance improvement for word sense disambiguation tasks. We then propose a sense alignment objective on top of the sense-aware cross entropy loss for cross-lingual model pretraining, and pretrain cross-lingual models for several language pairs (English to German/Spanish/Japanese/Chinese). Compared with the best baseline results, our cross-lingual models achieve 0.52%, 2.09% and 1.29% average performance improvements on zero-shot cross-lingual NER, sentiment classification and XNLI tasks, respectively. We will release our code.

## 1 INTRODUCTION

Cross-lingual word embeddings (CLWE) provide a shared representation space for knowledge transfer between languages, yielding state-of-the-art performance in many cross-lingual natural language processing (NLP) tasks. Most of the previous works have focused on aligning static embeddings. To utilize the richer information captured by the pre-trained language model, more recent approaches attempt to extend previous methods to align contextual representations.

Aligning the dynamic and complex contextual spaces poses significant challenges, so most of the existing approaches only perform coarse-grained alignment. Schuster et al. (2019) compute the average of contextual embeddings for each word as an anchor, and then learn to align the *static* anchors using a bilingual dictionary. In another work, Aldarmaki & Diab (2019) use parallel sentences in their approach, where they compute sentence representations by taking the average of contextual word embeddings, and then they learn a projection matrix to align sentence representations. They find that the learned projection matrix also works well for word-level NLP tasks. Besides, unsupervised multilingual language models (Devlin et al., 2018; Artetxe & Schwenk, 2019; Conneau et al., 2019; Liu et al., 2020) pretrained on multilingual corpora have also demonstrated strong cross-lingual transfer performance. Cao et al. (2020) and Wang et al. (2020) show that unsupervised multilingual language model can be further aligned with parallel sentences.

Though contextual word embeddings are intended to provide different representations of the same word in distinct contexts, Schuster et al. (2019) find that the contextual embeddings of different senses of one word are much closer compared with that of different words. This contributes to the anisomorphic embedding distribution of different languages and causes problems for cross-lingual alignment. For example, it will be difficult to align the English word *bank* and its Japanese translations 銀行 and 岸 that correspond to its two different senses, since the contextual embeddings of different senses of *bank* are close to each other while those of 銀行 and 岸 are far. Recently, Zhang et al. (2019) propose two solutions to handle multi-sense words: 1) remove multi-sense words and then align anchors in the same way as Schuster et al. (2019); 2) generate cluster level average anchor for contextual embeddings of multi-sense words and then learn a projection matrix in an unsupervised way with MUSE (Conneau et al., 2017). They do not make good use of the bilingual dictionaries,

which are usually easy to obtain, even in low-resource scenarios. Moreover, their projection-based approach still cannot handle the anisomorphic embedding distribution problem.

In this work, we propose a novel sense-aware cross entropy loss to model multiple word senses explicitly, and then leverage a sense level translation task on top of it for cross-lingual model pretraining. The proposed sense level translation task enables our models to provide more isomorphic and better aligned cross-lingual embeddings. We only use the cross-lingual signal from bilingual dictionaries for supervision. Our pretrained models demonstrate consistent performance improvements on zero-shot cross-lingual NER, sentiment classification and XNLI tasks. Though pretrained on less data, our model achieves the state-of-the-art result on zero-shot cross-lingual German NER task. To the best of our knowledge, we are the first to perform sense-level contextual embedding alignment with only bilingual dictionaries.

## 2 BACKGROUND: PREDICTION TASKS OF LANGUAGE MODELS

Next token prediction and masked token prediction are two common tasks in neural language model pretraining. We take two well-known language models, ELMo (Peters et al., 2018) and BERT (Devlin et al., 2018), as examples to illustrate these two tasks (architectures are shown in Appendix A).

**Next token prediction** ELMo uses next token prediction tasks in a bidirectional language model. Given a sequence of $N$ tokens $(t_1, t_2, \ldots, t_N)$, it first prepares a context independent representation for each token by using a convolutional neural network over the characters or by word embedding lookup (*a.k.a. input embeddings*). These representations are then fed into $L$ layers of LSTMs to generate the contextual representations: $\boldsymbol{h}_{i,j}$ for token $t_i$ at layer $j$. The model assigns a learnable *output embedding* $\boldsymbol{w}$ for each token in the vocabulary, which has the same dimension as $\boldsymbol{h}_{i,L}$. Then, the forward language model predicts the token at position $k$ with:

$$p(t_k|t_1, t_2, \ldots, t_{k-1}) = \text{softmax}(\boldsymbol{h}_{k-1,L}^{\mathsf{T}} \boldsymbol{w}_{k'}) = \frac{\exp(\boldsymbol{h}_{k-1,L}^{\mathsf{T}} \boldsymbol{w}_{k'})}{\sum_{i=1}^{V} \exp(\boldsymbol{h}_{k-1,L}^{\mathsf{T}} \boldsymbol{w}_i)} \tag{1}$$

where $k'$ is the index of token $t_k$ in the vocabulary, $V$ is the size of the vocabulary, and $(\boldsymbol{w}_1, \ldots, \boldsymbol{w}_V)$ are the output embeddings for the tokens in the vocabulary. The backward language model is similar to the forward one, except that tokens are predicted in the reverse order. Since the forward and backward language models are very similar, we will only describe our proposed approach in the context of the forward language model in the subsequent sections.

**Masked token prediction** The Masked Language Model (MLM) in BERT is a typical example of masked token prediction. Given a sequence $(t_1, t_2, \ldots, t_N)$, this approach randomly masks a certain percentage (15%) of the tokens and generates a masked sequence $(m_1, m_2, \ldots, m_N)$, where $m_k = [mask]$ if the token at position $k$ is masked, otherwise $m_k = t_k$. BERT first prepares the context independent representations $(\boldsymbol{x}_1, \boldsymbol{x}_2, \ldots, \boldsymbol{x}_N)$ of the masked sequence via token embeddings. It is then fed into $L$ layers of transformer encoder (Vaswani et al., 2017) to generate "bidirectional" contextual token representations. The final layer representations are then used to predict the masked token at position $k$ as follows:

$$p(m_k = t_k|m_1, \ldots, m_N) = \text{softmax}(\boldsymbol{h}_{k,L}^{\mathsf{T}} \boldsymbol{w}_{k'}) = \frac{\exp(\boldsymbol{h}_{k,L}^{\mathsf{T}} \boldsymbol{w}_{k'})}{\sum_{i=1}^{V} \exp(\boldsymbol{h}_{k,L}^{\mathsf{T}} \boldsymbol{w}_i)} \tag{2}$$

where $k'$, $V$, $\boldsymbol{h}$ and $\boldsymbol{w}$ are similarly defined as in Eq. 1. Unlike ELMo, BERT ties the input and output embeddings.

## 3 PROPOSED FRAMEWORK

We first describe our proposed sense-aware cross entropy loss to model multiple word senses explicitly in language model pretraining. Then, we present our joint training approach with sense alignment objective for cross-lingual mapping of contextual word embeddings. The proposed framework can be applied to most of the recent neural language models, such as ELMo, BERT and their variants. See Table 1 for a summary of the main notations used in this paper.

### 3.1 SENSE-AWARE CROSS ENTROPY LOSS

**Limitations of original training objectives**
The training tasks with Eq. 1 and 2 maximize the normalized dot product of contextual representations ($h_{k-1,L}$ or $h_{k,L}$) with a weight vector $w_{k'}$. The only difference is that $h_{k-1,L}$ in Eq. 1 encodes the information of previous tokens in the sequence, while $h_{k,L}$ in Eq. 2 encodes the information of the masked sequence. Therefore, without loss of generality, we use $h_{k^*,L}$ to denote the contextual representation for predicting the next or masked token $t_k$.

Even though contextual language models like ELMo and BERT provide a different token representation for each distinct context, the learned representations are not guaranteed to be sense separated. For example, Schuster et al. (2019) computed the average of ELMo embeddings for each word as an *anchor*, and found that the average cosine distance between contextual em-

Table 1: Summary of the main notations

| Notation | Description |
| --- | --- |
| $t_k$ | $k$-th token in sentence |
| $t_{k,s}$ | $s$-th sense of $t_k$ |
| $k'$ | index of token $t_k$ in vocabulary |
| $L$ | number of LSTM/Transformer layers |
| $V$ | size of vocabulary |
| $S$ | maximum number of senses per token |
| $h_{k,j}$ | contextual representation of token $t_k$ in layer $j$ |
| $h_{k^*,L}$ | contextual representation used in softmax function for predicting $t_k$ |
| $v_i$ | $i$-th word in vocabulary |
| $v_{i,s}$ | $s$-th sense of $v_i$ |
| $w_i$ | output embedding of $v_i$ |
| $w_{i,s}$ | context-dependent output embedding (i.e. sense vector) of $v_{i,s}$ |
| $c_{i,s}$ | sense cluster center of $v_{i,s}$ |
| $C_i$ | sense cluster centers of $v_i$ |
| $d$ | dimension of contextual representations |
| $P$ | projection matrix for dimension reduction |

beddings of multi-sense words and their corresponding anchors are much smaller than the average distance between anchors, which mean that the embeddings of different senses of one word are relatively near to each other comparing to that of different words. We also observed the same with BERT embeddings. This finding suggests that sense clusters of a multi-sense word's appearances are not well separated in the embedding space, and the current contextual language models still have room for improvement by considering finer-grained word sense disambiguation.

Notice that there is only one weight vector $w_{k'}$ for predicting the token $t_k$ in the original training tasks. Ideally, we should treat the appearances of a multi-sense word in different contexts as different tokens, and train the language models to predict different senses of the word. In the following, we propose a novel sense-aware cross entropy loss to explicitly model different senses of a word in different contexts.

**Sense-aware cross entropy loss**  Given a sequence $(t_1, t_2, \ldots, t_N)$, our proposed framework generates contextual representations ($h_{k,j}$ for token $t_k$ in layer $j \in \{1, \ldots, L\}$) in the same way as the standard LMs. Different from existing methods, our approach maintains multiple *context-dependent output embeddings* (henceforth, sense vectors) for each token. Specifically, let $S$ be the maximum number of senses per token. Each word $v_i$ in the vocabulary contains $S$ separate sense vectors $(w_{i,1}, w_{i,2}, \ldots, w_{i,S})$, where each $w_{i,s}$ corresponds to a different sense (see Appendix for some interesting visualization examples). Following the notation in Section 2, we use $k'$ to denote the index of the output token $t_k$ in the vocabulary. Therefore, the sense vectors of $t_k$ can be represented by $(w_{k',1}, w_{k',2}, \ldots, w_{k',S})$, which are randomly initialized and of the same dimension as $h_{k^*,L}$. Note that we untie the input and output embeddings in our framework.

We propose a word sense selection method shown in Algorithm 1 to select the most likely sense vector when training with sense-level cross entropy loss. Figure 1 shows the architecture of our proposed models. Assuming sense $s'$ is selected for token $t_k$ (which means sense vector $w_{k',s'}$ should be used), we have the following new prediction task:

$$p(t_{k,s'}|context) = \text{softmax}(h_{k^*,L}^\mathsf{T} w_{k',s'}) = \frac{\exp(h_{k^*,L}^\mathsf{T} w_{k',s'})}{\sum_{i=1}^{V} \sum_{s=1}^{S} \exp(h_{k^*,L}^\mathsf{T} w_{i,s})} \tag{3}$$

The sense-aware cross entropy loss for word sense prediction is defined as follows:

$$\mathcal{L}_{\text{SENSE}} = -\log(p(t_{k,s'}|context)) \tag{4}$$

**Word sense selection algorithm**  Word sense selection when training the language model can be handled as a non-stationary data stream clustering problem (Aggarwal et al., 2004; Khalilian & Mustapha, 2010; Abdullatif et al., 2018). The most intuitive way to select the corresponding sense

---

[1]Since the backward language model is similar to the forward, we only show the forward one for simplicity.

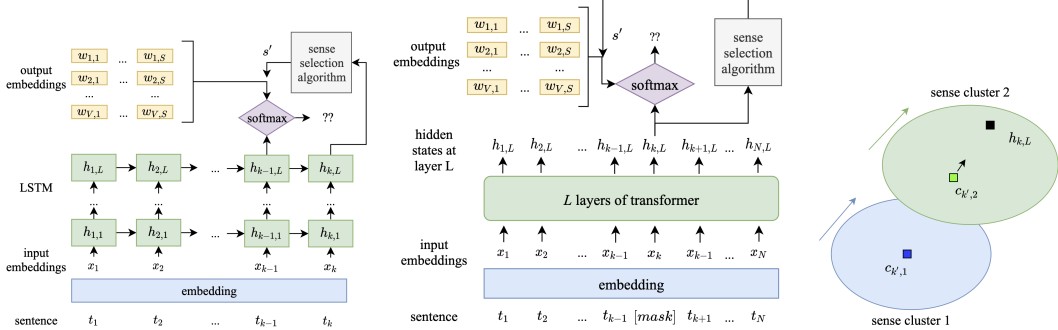

(a) Sense-aware next token prediction    (b) Sense-aware masked token prediction    (c) Word sense selection

Figure 1: Our proposed framework for sense-aware next token[1] and masked token prediction tasks. Figure (c) shows an example of word sense selection, where the two sense clusters of $t_k$ (assume its vocabulary index is $k'$) are shifting in space. Center vectors $c_{k',1}$ and $c_{k',2}$ are used to locate cluster centers. Given $h_{k,L}$, the algorithm performs dimension reduction on both $h_{k,L}$ and center vectors, and then finds the most close cluster center $c_{k',2}$, so we know the output embedding corresponding to sense 2 ($w_{k',2}$) should be used in the loss function. $c_{k',2}$ also makes a small step towards $h_{k,L}$.

vector for $h_{k^*,L}$ is to select the vector $w_{k',s}$ with the maximum dot product value $h_{k^*,L}^{\mathsf{T}} w_{k',s}$, or cosine similarity value $cossim(h_{k^*,L}, w_{k',s})$. However, our experiments show that these methods do not work well due to curse of dimensionality, suboptimal learning rate and noisy $h_{k^*,L}$. We apply an online k-means algorithm to cluster different senses of a word in Algorithm 1. For each sense vector $w_{i,s}$, we maintain a cluster *center* $c_{i,s}$ which is of the same dimension as $w_{i,s}$. Therefore, each token $v_i$ in the vocabulary has $S$ such cluster center vectors, denoted by $C_i = (c_{i,1}, c_{i,2}, \ldots, c_{i,S})$. When predicting token $t_k$ in a given sequence, we apply Algorithm 1 to select the best sense vector based on $h_{k,L}$ (see Figure 1). Notice that $h_{k,L}$ is different from $h_{k^*,L}$ for next token prediction (Figure 1a) for which $h_{k^*,L} = h_{k-1,L}$. The cluster centers $C_i$ are **not** neural network parameters; instead, they are randomly initialized using a normal distribution $\mathcal{N}(0, \sigma^2)$ and updated through Algorithm 1. In addition, we also maintain a projection matrix $P$ for dimension reduction to facilitate effective sense clustering. $P \in \mathbb{R}^{d \times d'}$ projects $h_{k,L}$ and $c_{i,s}$ from dimension $d$ to $d'$, and is shared by all tokens in vocabulary. Similar to $C$, $P$ is also randomly initialized with normal distribution $\mathcal{N}(0,1)$, and then updated through Algorithm 2. Both Algorithm 1 and 2 run in parallel, and are interrupted when the language model stops training.

Some rationales behind our algorithm design are the following:

---

**Algorithm 1** Word sense selection

1: **Hyper-parameters:** number of senses $S$, sense learning rate $\alpha$
2: Initialize the set of all sense cluster centers $C$
3: **repeat**
4:     **input:** $h_{k,L}$, vocabulary index $k'$ of the token to predict
5:     Lookup sense cluster centers for $k'$: $C_{k'} = \{c_{k',1}, c_{k',2}, \ldots, c_{k',S}\}$
6:     $P$ = updated projection matrix from Alg. 2
7:     **if** cosine similarity between $c_{k',s'}P$ and $h'_k P$ is the largest among the vectors in $C_{k'}$ **then**
8:        $c_{k',s'} = (1-\alpha)c_{k',s'} + \alpha h_{k,L}$
9:        **output:** $s'$ ($w_{k',s'}$ should be selected)
10:    **end if**
11: **until** interrupted

---

**Algorithm 2** Projection matrix $P$ update

1: **Hyper-parameters:** projection dimension $d'$, update interval $M$, queue size $Q$
2: Initialize $P$ with $\mathcal{N}(0,1)$, queue $H = \emptyset$, $m = 0$
3: **repeat**
4:     **input:** $h_{k,L}$
5:     $m = m + 1$
6:     Add $h_{k,L}$ to queue $H$
7:     **if** $size(H) > Q$ **then**
8:        Pop the oldest element from queue $H$.
9:     **end if**
10:    **if** $m >= M$ **then**
11:       $P$ = the first $d'$ PCA components of $H$
12:       $m = 0$
13:    **end if**
14:    **output:** $P$
15: **until** interrupted

- Directly computing cosine similarity between $c_{k',s}$ and $h_{k,L}$ suffers from the curse of dimensionality. We maintain $P$ for dimension reduction. Although many algorithms use random projection for dimension reduction, we find using PCA components can help improve clustering accuracy.

- Since the neural model parameters keep being updated during training, the sense clusters become non-stationary, i.e., their locations keep changing. Experiments shows that when using $P$ for dimension reduction, a slightly larger projection dimension $d'$ will make the clustering algorithm less sensitive to cluster location change. We use $d' = 16$ for ELMo, and $d' = 14$ for BERT. We also notice that the sense clustering works well even if $P$ is updated sporadically. We can set a relatively large update interval in Algorithm 2 to reduce computation cost.

- A separate sense learning rate $\alpha$ should be set for the clustering algorithm. A large $\alpha$ makes the algorithm less robust to noise, while a small $\alpha$ leads to slow convergence.

- It is essential to use the current token's contextual representation $h_{k,L}$ for sense selection even though we use $h_{k^*,L} = h_{k-1,L}$ in the next token prediction task. If we use $h_{k-1,L}$ for sense selection, experiments show that most of the variance comes from input embedding $x_{k-1}$. This introduces too much noise for word sense clustering.

**Dynamic pruning of redundant word senses**    To make the training more efficient, we keep track of relative sense selection frequency for each token in the vocabulary. Assume token $v_i$ has initial senses $(v_{i,1}, v_{i,2}, \ldots, v_{i,S})$, for which we compute the relative frequency $\rho(v_{i,s})$ such that $0 \leq \rho(v_{i,s}) \leq 1$ and $\sum_s \rho(v_{i,s}) = 1$. A lower $\rho(v_{i,s})$ means the sense is less frequently selected compared with others. We check the relative frequencies after every $E$ training steps, and if $\rho(v_{i,s}) < \beta$ (a threshold hyper-parameter), $v_{i,s}$ is removed from the list of senses of $v_i$.

**Remark on model size and parameters**    The sense cluster centers $C$ and the projection matrix $P$ are only used to facilitate sense selection during model pretraining, which are not neural model parameters. The sense vectors $w_{i,s}$ will no longer be used after pretraining, which can also be discarded. Therefore, our models and the original models have exactly the same number of parameters when transferred to downstream tasks.

**Remark on model complexity**    The computational complexity of our algorithm is linear with respect to the size of data, so our method is scalable to train on very large datasets.

### 3.2   JOINT TRAINING WITH SENSE LEVEL TRANSLATION

Training language model with sense-aware cross entropy loss helps to learn contextual token representations that are sufficiently distinct for different senses (§4.1). In this subsection, we extend it to cross-lingual settings and present a novel approach to learn cross-lingual contextual word embeddings at the sense level. Our approach uses a bilingual seed dictionary,[2] and can be applied to both next and masked token prediction tasks.

For training the cross-lingual LM, we concatenate the (non-parallel) corpora of two languages, $L_1$ and $L_2$, and construct a joint vocabulary $O = O^{L_1} \cup O^{L_2}$, where $O^{L_1}$ and $O^{L_2}$ are the vocabularies of $L_1$ and $L_2$, respectively. Algorithm 1 is used to model the senses of tokens in the joint vocabulary. In addition to predicting the correct monolingual sense $p(t_{k,s'}|context)$ in Eq. 3, we also train the model to predict its sense level translation. Let $v_j$ be the translation of $t_k$ and sense $v_{j,s^*}$ of $v_j$ be the best sense level translation under the given context, we add the following sense-level translation prediction task to maximize probability of $v_{j,s^*}$.

$$p(v_{j,s^*}|context) = \text{softmax}(h_{k^*,L}^\mathsf{T} w_{j,s^*}) = \frac{\exp(h_{k^*,L}^\mathsf{T} w_{j,s^*})}{\sum_{i=1}^{V} \sum_{s=1}^{S} \exp(h_{k^*,L}^\mathsf{T} w_{i,s})} \tag{5}$$

where $w_{j,s^*}$ is the corresponding sense vector of $v_{j,s^*}$.

Similar to the previous subsection, we maintain sense cluster centers $C_i$ for each token $v_i \in O$ and the shared projection matrix $P$ to select the best translation sense. Assume $t_k$ has $T$ translations in dictionary, and each translation has $S$ senses, then there are $T \times S$ possible sense level translations for $t_k$ in the given context. If the $cossim(h_{k,L}P, c_{j,s^*}P)$ value is the largest among the $T \times S$ sense cluster centers, then we select $v_{j,s^*}$ as the closest translation. An example is shown in Figure 2.

---

[2]If not provided, it can be learned in an unsupervised way, e.g., MUSE (Conneau et al., 2017).

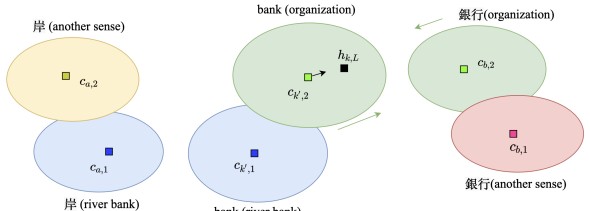

Figure 2: An example of English-Japanese sense-level joint training, which shows two possible Japanese translations (銀行 and 岸) of the English word *bank*. $h_{k,L}$ is a contextual representation of *bank* in finance context and $c_{k',2}$ is the cluster center for this sense. $c_{a,1}$, $c_{a,2}$, $c_{b,1}$, $c_{b,2}$ are different sense cluster centers of the two Japanese translations, among which $c_{b,2}$ is the closest to $h_{k,L}$ after dimension reduction through PCA. Our sense level objective (Eq. 6) moves sense clusters for *bank (organization)* and 銀行*(organization)* closer to each other.

If token $t_k$ has at least one translation in the dictionary, the translation cross entropy loss can be computed as:

$$\mathcal{L}_{\text{TRAN}} = -\log(p(v_{j,s^*}|context)) \tag{6}$$

If token $t_k$ has no translation in the seed dictionary, we use Eq. 4 as the only loss. The joint training loss is defined as follows:

$$\mathcal{L}_{\text{JOINT}} = \begin{cases} \frac{\mathcal{L}_{\text{SENSE}} + \mathcal{L}_{\text{TRAN}}}{2}, & \text{if } t_k \text{ has translations} \\ \mathcal{L}_{\text{SENSE}}, & \text{otherwise} \end{cases} \tag{7}$$

**Further alignment (optional)** Our sense-aware pretraining tries to move similar senses of two different languages close to each other as illustrated in Figure 2. This process makes the sense distributions of the two languages more isomorphic (some sense vector visualization examples are shown in Appendix C). Applying the linear projection approach proposed by Schuster et al. (2019) on top of the language model pretrained with our framework can further improve cross-lingual transfer on some tasks. See Appendix B for more details of our implementation.

## 4 EXPERIMENTS

### 4.1 EXPERIMENTS USING MONOLINGUAL MODELS

To verify the effectiveness of our proposed sense-aware cross entropy loss, we implement the monolingual models on top of ELMo and BERT with the changes described in §3.1, which are named **SaELMo** (Sense-aware ELMo) and **SaBERT** (Sense-aware BERT) respectively. The algorithm for dynamic pruning of redundant word senses is optional, which is implemented on SaELMo only.

**Pretraining settings** We use the one billion word language modeling benchmark data (Chelba et al., 2013) to pretrain all the monolingual models. The corpus is preprocessed with the provided scripts, and then converted to lowercase. We do not apply any subword tokenization. We use similar hyper-parameters as Peters et al. (2018) to train the ELMo and SaELMo models, and similar hyper-parameters as Devlin et al. (2018) to train 4-layer BERT-Tiny and SaBERT-Tiny. Next sentence prediction task is disabled in BERT-Tiny and SaBERT-Tiny, since this task is irrelevant to our proposed changes. See Appendix D.1 for a complete list of hyper-parameters.

**Word sense disambiguation (WSD)** Since our context-aware cross entropy loss is designed to learn word senses better in the context, we first conduct experiments to compare our monolingual model with the original models on the WSD task (Raganato et al., 2017), which is a task to associate words in context with the most suitable entry in a pre-defined sense inventory. We use a similar framework as Peters et al. (2018) to evaluate the monolingual models.[3] We use SemCor 3.0 (Miller et al., 1993) as training data, and Senseval/SemEval series (Edmonds & Cotton, 2001; Moro & Navigli, 2015; Navigli et al., 2013; Pradhan et al., 2007; Snyder & Palmer, 2004) as test data. We use the pretrained models to compute the average of contextual representations for each sense in training data, and then classify the senses of the target words in test sentences by finding the nearest neighbour.

---

[3]We modified the script from: https://github.com/drgriffis/ELMo-WSD-reimplementation.git

WSD results are presented in Table 2. SaELMo shows significant performance improvements over the baseline ELMo model in all of the five test sets. SaBERT-Tiny also outperforms BERT-Tiny except on SE07, which is the smallest among the five test sets.

Table 2: Word sense disambiguation (F1 scores)

| Model | SE2 | SE3 | SE07 | SE13 | SE15 |
|---|---|---|---|---|---|
| ELMo | 0.555 | 0.576 | 0.446 | 0.544 | 0.538 |
| SaELMo (ours) | **0.575** | **0.586** | **0.470** | **0.560** | **0.583** |
| BERT-Tiny | 0.596 | 0.539 | **0.466** | 0.536 | 0.572 |
| SaBERT-Tiny (ours) | **0.611** | **0.546** | 0.446 | **0.550** | **0.579** |

## 4.2 EXPERIMENTS USING BILINGUAL MODELS

To verify the effectiveness of our cross-lingual framework, we implement the bilingual models on top of ELMo, named **Bi-SaELMo** that does not use linear projection for further alignment and **Bi-SaELMo+Proj** that uses the linear projection. Sense vectors and cluster center vectors are not shared between the forward and backward language models. We use **ELMo+Proj** and **Joint-ELMo+Proj** as our baseline models, where ELMo+Proj is proposed by Schuster et al. (2019) and Joint-ELMo+Proj is implemented following the framework recently proposed by Wang et al. (2020). Wang et al. (2020) combine joint training and projection, and claim their framework is applicable to any projection method, so we implement the same projection method as Schuster et al. (2019) did for Joint-ELMo+Proj. We also report results of **ELMo** and **Joint-ELMo**, which are the counterparts of ELMo+Proj and Joint-ELMo+Proj without using linear projection.

**Pretraining settings** To pretrain language models, we sample a 500-million-token corpus for each language from the English, German, Spanish, Japanese and Chinese Wikipedia dump. The dictionaries used for pretraining models and learning the projection matrix were downloaded from the MUSE (Conneau et al., 2017) GitHub page[4]. We also add JMDict (Breen, 2004) to the *en-jp* MUSE dictionary. Bilingual models were pretrained on *en-de*, *en-es*, *en-jp* and *en-zh* concatenated data with similar parameters as the monolingual models. ELMo and ELMo+Proj were pretrained on monolingual data, while the projection matrix of ELMo+Proj was learned using bilingual data. See Appendix D.2 for a complete list of hyper-parameters.

**Zero-shot cross-lingual NER** A Bi-LSTM-CRF model implemented with the Flair framework (Akbik et al., 2018) is used for this task. For the CoNLL-2002 (Tjong Kim Sang, 2002) and CoNLL-2003 (Sang & De Meulder, 2003) datasets, the NER model was trained on English data, and evaluated on Spanish and German test data. For the OntoNotes 5.0 (Weischedel et al., 2013) dataset, the NER model was trained on all English data and evaluated on all Chinese data. We report the average F1 of 5 runs in Table 3. The results show that all of the models using linear projection outperform their counterparts (not using linear projection), since minimizing token level distance is more important for cross-lingual

Table 3: Zero-shot cross-lingual NER (F1)

| Model | de | es | zh |
|---|---|---|---|
| ELMo | 16.30 | 16.14 | 0.28 |
| Joint-ELMo | 56.49 | 58.91 | 53.47 |
| ELMo+Proj (Schuster et al., 2019) | 69.57 | 60.02 | 63.15 |
| Joint-ELMo+Proj (Wang et al., 2020) | 71.59 | 65.19 | 59.08 |
| Bi-SaELMo (ours) | 63.83 | 60.65 | 55.83 |
| Bi-SaELMo+Proj (ours) | **72.19** | **65.86** | **63.44** |
| **For references**, but not our baselines, since they are trained on much larger datasets and/or parallel sentences. | | | |
| XLM Finetune (Conneau & Lample, 2019) | 67.55 | 63.18 | - |
| XLM-R Finetune (Conneau et al., 2019) | 71.40 | 78.64 | - |
| M-BERT Finetune (Pires et al., 2019) | 69.74 | 73.59 | - |
| M-BERT Finetune (Wu & Dredze, 2019) | 69.56 | 74.96 | - |
| M-BERT Finetune+Adv (Keung et al., 2019) | 71.90 | 74.30 | - |
| M-BERT Feature+Proj (Wang et al., 2020) | 70.54 | 75.77 | - |

NER tasks. Our sense-aware pretraining makes sense distributions of two languages more isomorphic, which further improves linear projection performance. Our model Bi-SaELMo+Proj demonstrates consistent performance improvement in all the three languages. Moreover, our model outperforms finetuned XLM/XLM-R and Multilingual BERT on German data, and achieves state of the art even though it is pretrained on less data.

---

[4]https://github.com/facebookresearch/MUSE

**Zero-shot cross-lingual sentiment classification** We use the multi-lingual multi-domain Amazon review data (Prettenhofer & Stein, 2010) for evaluation on cross-lingual sentiment classification. The ratings in review data are converted into binary labels. The average of contextual word representations is used as the document/sentence representation for each review text/summary, which is then fed into a two-dense-layer model for sentiment classification. All the models are trained on English, and evaluated on German and Japanese test data in the same domain. We report the average accuracy of 5 runs in Table 4. Different from the NER task, the linear projection approach for cross-lingual alignment does not work for this task, since it may add noise to embedding features. Our model Bi-SaELMo demonstrates consistent improvements in all of the 6 evaluation tasks. The performance of Bi-SaELMo is significantly better than Joint-ELMo, which shows that our sense-level translation pretraining objective improves cross-lingual embedding alignment.

Table 4: Zero-shot sentiment classification accuracy

| Model | de | | | jp | | |
|---|---|---|---|---|---|---|
| | books | music | dvd | books | music | dvd |
| ELMo | 52.94 | 63.61 | 57.78 | 50.37 | 51.59 | 54.32 |
| Joint-ELMo | 71.72 | 75.22 | 64.25 | 66.64 | 68.50 | 58.54 |
| ELMo+Proj (Schuster et al., 2019) | 49.92 | 50.29 | 49.94 | 50.57 | 49.59 | 50.65 |
| Joint-ELMo+Proj (Wang et al., 2020) | 75.74 | 72.25 | 72.25 | 62.50 | 59.77 | 57.65 |
| Bi-SaELMo (ours) | **77.46** | **75.32** | **74.97** | **68.16** | **69.48** | **64.04** |
| Bi-SaELMo+Proj (ours) | 70.84 | 66.25 | 68.99 | 62.17 | 55.91 | 61.57 |

Table 5: Zero-shot XNLI accuracy

| Model | de | es | zh |
|---|---|---|---|
| ELMo | 34.07 | 33.41 | 35.77 |
| Joint-ELMo | 60.12 | 63.73 | 57.82 |
| ELMo+Proj (Schuster et al., 2019) | 55.51 | 58.92 | 53.17 |
| Joint-ELMo+Proj (Wang et al., 2020) | 63.33 | 64.71 | 58.34 |
| Bi-SaELMo (ours) | 60.98 | 62.75 | 60.40 |
| Bi-SaELMo+Proj (ours) | **64.77** | **65.05** | **60.44** |

**Zero-shot cross-lingual natural language inference (XNLI)** We use XNLI (Conneau et al., 2018) and MultiNLI (Williams et al., 2018) data for evaluation on this task. The Bi-LSTM baseline model[5] was trained on MultiNLI English training data, and then evaluated on XNLI German, Spanish, Chinese test data. We report the average zero-shot XNLI accuracy of 2 runs in Table 5. Our models show consistent improvements over the baselines on all of the three data sets. For zero-shot transfer to Chinese, both of our models outperform the best baseline by more than 2 points, which again demonstrates the effectiveness of our framework on distant language pairs.

## 5 RELATED WORK

Cross-lingual word embedding demonstrates strong performance in many cross-lingual transfer tasks. The projection-based approach has a long line of research on aligning static embeddings (Mikolov et al., 2013; Xing et al., 2015; Smith et al., 2017; Joulin et al., 2018). It assumes that the embedding spaces of different languages have an isomorphic structure, and fit an orthogonal matrix to project multiple monolingual embedding spaces to a shared space. Recent studies (Schuster et al., 2019; Aldarmaki & Diab, 2019) have extended this approach to contextual representation alignment. Besides, there are also many discussions on the limitations of the projection-based approach, arguing that the isomorphic assumption is not true in general (Nakashole & Flauger, 2018; Patra et al., 2018; Søgaard et al., 2018; Ormazabal et al., 2019). Joint training is another line of research and early methods (Gouws et al., 2015; Luong et al., 2015; Ammar et al., 2016) learn static word embeddings of multiple languages simultaneously. Extending joint training to cross- or multi-lingual language model pretraining has gained more attention recently. As discussed above, unsupervised multilingual language models (Devlin et al., 2018; Artetxe & Schwenk, 2019; Conneau & Lample, 2019; Conneau et al., 2019; Liu et al., 2020) also demonstrate strong cross-lingual transfer performance.

There has been some work on sense-aware language models/embeddings (Rothe & Schütze, 2015; Pilehvar & Collier, 2016; Hedderich et al., 2019), and most of them require WordNet (Miller, 1998) or other additional resource for supervision. Šuster et al. (2016) utilize both monolingual and bilingual information from parallel corpora to learn multi-sense word embeddings. Peters et al. (2019) embed WordNet knowledge into BERT with attention mechanism. Levine et al. (2019) pretrain SenseBERT to predict both the masked words and their WordNet supersenses. Similar to our framework, there are also some unsupervised approaches, but most of them are used to learn static embeddings. Huang et al. (2012) learn word representations with both local and global context, and then apply a clustering algorithm to learn multi-prototype vectors. Neelakantan et al. (2014) propose an extension to the Skip-gram model that leverage k-means clustering algorithm learns multiple embeddings per word type. Lee & Chen (2017) leverage reinforcement learning for modularized unsupervised sense level embedding learning. Boyd-Graber et al. (2020) use Gumbel softmax for sense disambiguation when learning sense embeddings.

---

[5]https://github.com/NYU-MLL/multiNLI

## 6 CONCLUSIONS

In this paper, we have introduced a novel sense-aware cross entropy loss to model word senses explicitly, then we have further proposed a sense-level alignment objective for cross-lingual model pretraining using only bilingual dictionaries. The results of the experiments show the effectiveness of our monolingual and bilingual models on WSD, zero-shot cross-lingual NER, sentiment classification and XNLI tasks. In future work, we will study how to effectively extend our method to multilingual models. In addition, using the sense cluster centers to learn the linear projection matrix would be another promising direction to further improve cross-lingual alignment.

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

APPENDIX

## A   PREDICTION TASKS OF LANGUAGE MODELS

Next token prediction and masked token prediction are two common tasks in neural language model (LM) pretraining. We take two well-known language models, ELMo and BERT, as examples to illustrate these two tasks, which are shown in Figure 3.

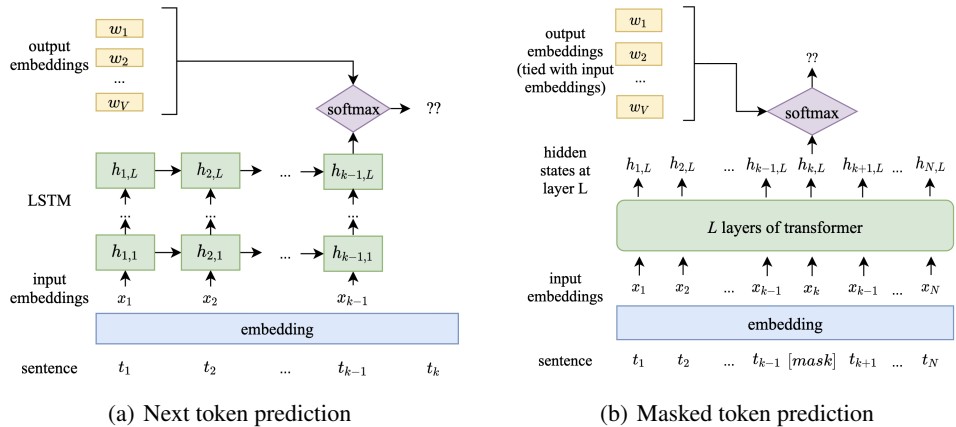

(a) Next token prediction         (b) Masked token prediction

Figure 3: Next token and masked token prediction tasks of language models. For simplicity, we only show the forward language model in next token prediction.

## B   FURTHER ALIGNMENT (OPTIONAL)

Applying the linear projection approach proposed by Schuster et al. (2019) on top of our framework can further improve cross-lingual transfer on some tasks. After our cross-lingual model is finished training on the concatenated corpora of two languages, $L_1$ and $L_2$, it is used to generate contextual token embeddings for the word pairs in the seed dictionary $\mathcal{D} = \{(t_i^{L_1}, t_i^{L_2})\}_{i=1}^{|\mathcal{D}|}$ [6]. Then, we compute the average of all contextual embeddings for each token $t_i^{L_j}$, denoted by $\boldsymbol{a}_i^{L_j}$. Finally, a linear projection matrix $\boldsymbol{W} \in \mathbb{R}^{d \times d}$ is learned to minimize cross-lingual embedding distance:

$$\boldsymbol{W} = \arg\min_{\boldsymbol{W}} \sum_{i=1}^{|\mathcal{D}|} ||\boldsymbol{W}\boldsymbol{a}_i^{L_1} - \boldsymbol{a}_i^{L_2}||^2 \qquad (8)$$

## C   VISUALIZATION OF SENSE VECTORS

We visualize[7] the sense vectors of each model in a two dimensional PCA, and show some examples in Figures 4 to 7. For our English monolingual model (SaELMo), the vectors close to two different sense vectors of the word *may* are shown in (a) and (b) of Figure 4, respectively. We observe that senses are well clustered in these two subfigures, where cluster (a) corresponds to "month", and cluster (b) corresponds to "auxiliary verb".

We do the same for the English-Japanese bilingual model (Bi-SaELMo, without projection), and show the vectors close to two different sense vectors of the English word *bank* in (c) and (d) of Figure 5. We can see both English and Japanese sense vectors (*trade*, 銀行, 証券, etc.) in (c), most of which correspond to the sense "organization", though there are some noises. Similarly, most of the sense vectors in (d) correspond to sense "river bank".

---

[6]If any token $t_k$ appears in both languages, we add that as an entry $(t_k, t_k)$ to the dictionary as well.

[7]We use the tensorflow embedding projector (https://projector.tensorflow.org/) for visualization.

Another two examples are shown in Figures 6 and 7. Our framework exhibits good sense clustering and sense level cross-lingual alignment behaviour in these examples. All sense vectors are dumped at training step 200,000, which is before pretraining complete.

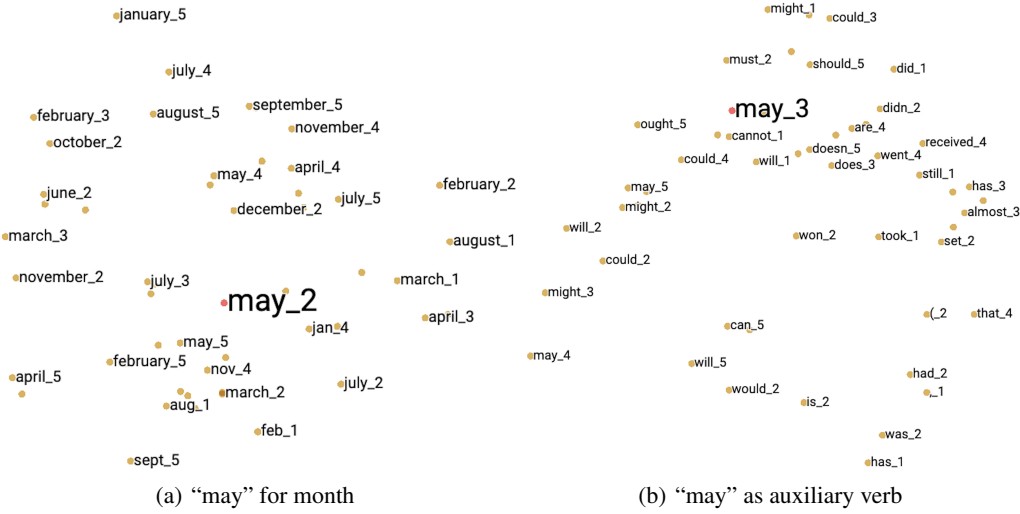

(a) "may" for month  (b) "may" as auxiliary verb

Figure 4: We visualize sense vectors of English monolingual model (SaELMo) in a two dimensional PCA, and show the vectors close to two different sense vectors of word *may* in (a) and (b).

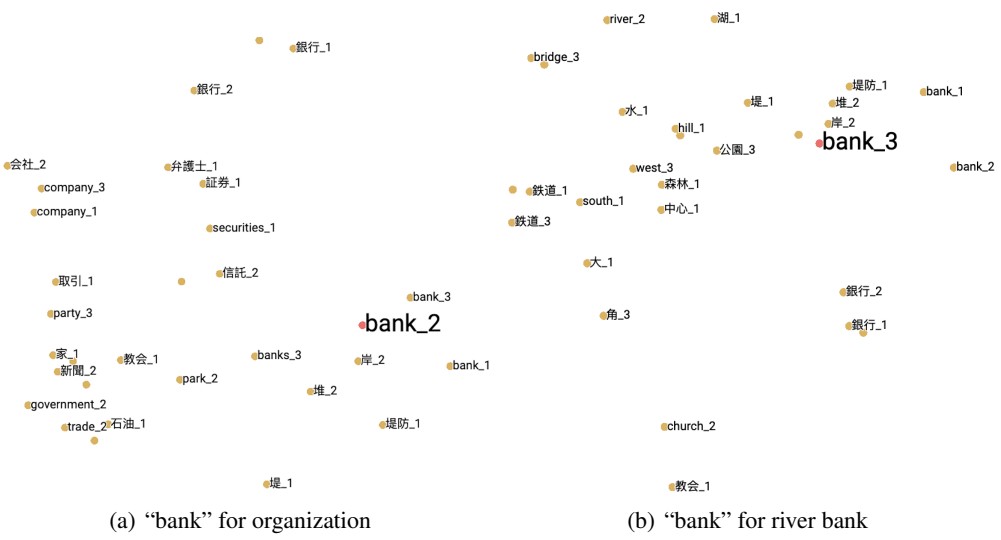

(a) "bank" for organization  (b) "bank" for river bank

Figure 5: We visualize all sense vectors of *en-jp* bilingual model (Bi-SaELMo) in a two dimensional PCA, and show the vectors close to two different sense vectors of word *bank*.

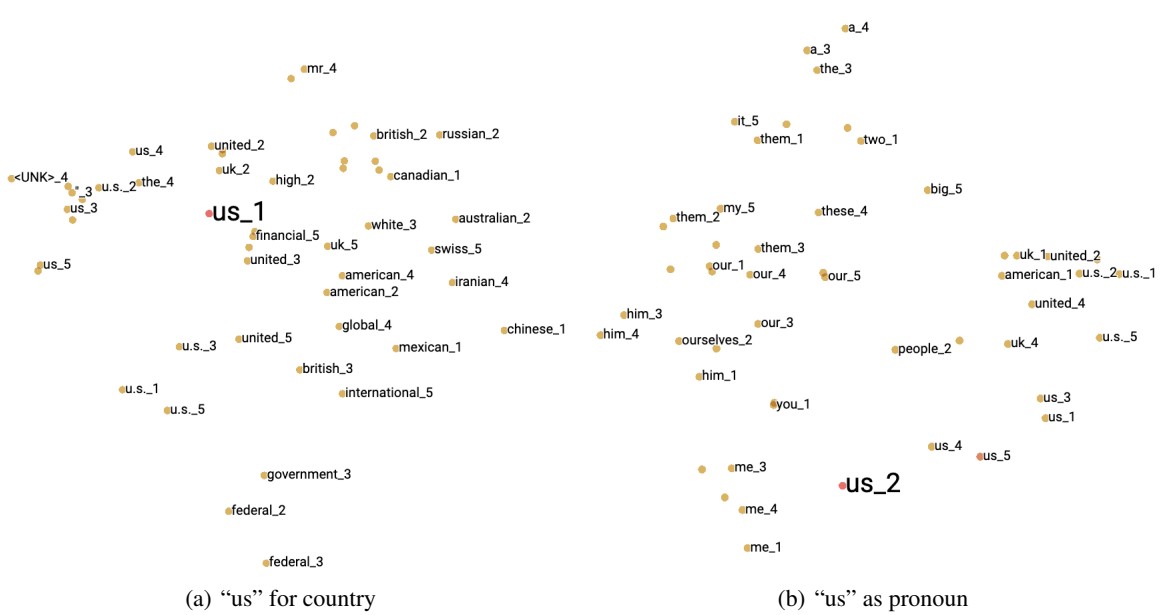

(a) "us" for country                (b) "us" as pronoun

Figure 6: We visualize sense vectors of English monolingual model (SaELMo) in a two dimensional PCA, and show the vectors close to two different sense vectors of word *us* in (a) and (b).

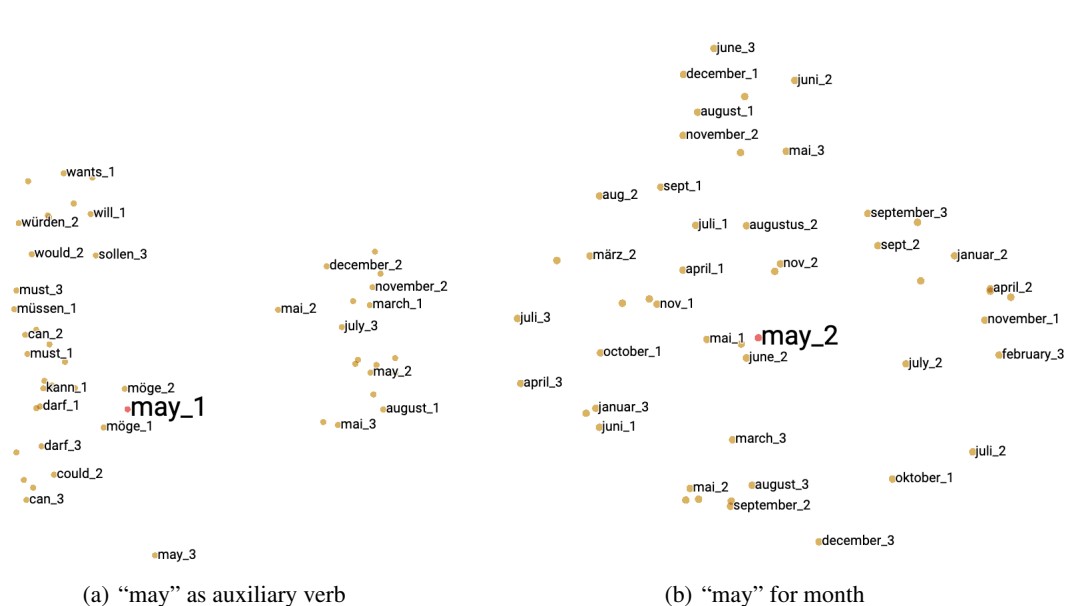

(a) "may" as auxiliary verb          (b) "may" for month

Figure 7: We visualize all sense vectors of *en-de* bilingual model (Bi-SaELMo) in a two dimensional PCA, and show the vectors close to two different sense vectors of word *may*.

## D   PRETRAINING DETAILS

### D.1   MONOLINGUAL MODEL

All the monolingual models were trained for one million steps. For better sense clustering performance, the maximum number of senses ($S$ in word sense selection algorithm) was set to 1 for the first 20,000 steps to quickly get a reasonable initial model, and then increased to 5 afterwards when pretraining SaELMo and SaBERT-Tiny, which is controlled by hyperparameter *n_context* in our implementation. For SaELMo, we set *n_context* to 6, so that the model initialize 6 senses for each token, but only use the first sense in the 20,000 steps, and then use the other 5 senses (the first sense will be disabled) afterwards. We implement this for SaBERT-Tiny in a slightly different way, where *n_context* can be set to 5 directly to achieve the same effect. We use two NVIDIA V100 GPUs to pretrain SaELMo, which takes about 15 days to complete training. We use one NVIDIA V100 GPU to pretrain SaBERT-Tiny, which takes about 5 days. See Tables 6 and 7 for the hyperparameters used to pretrain SaELMo and SaBERT-Tiny respectively.

Table 6: Monolingual model hyperparameters: SaELMo

| Hyperparameter | Value |
|---|---|
| max_word_length | 50 |
| batch_size | 256 |
| n_gpus | 2 |
| bidirectional | True |
| char_cnn:embedding:dim | 16 |
| char_cnn:max_characters_per_token | 50 |
| char_cnn:n_characters | 261 |
| char_cnn:n_highway | 2 |
| dropout | 0.1 |
| lstm:cell_clip | 3 |
| lstm:dim | 4096 |
| lstm:n_layers | 2 |
| lstm:proj_clip | 3 |
| lstm:projection_dim | 512 |
| lstm:use_skip_connections | True |
| all_clip_norm_val | 10.0 |
| n_epochs | 10 |
| unroll_steps | 16 |
| n_negative_samples_batch | 8192 |
| n_context | 6 |
| cluster_proj_dim | 16 |
| pca_sample | 20,000 |
| remove_less_freqent_contexts | 0.1 |
| learning_rate | 0.2 |
| sense_learning_rate | 0.01 |

Table 7: Monolingual model hyperparameters: SaBERT-Tiny

| Hyperparameter | Value |
|---|---|
| attention_probs_dropout_prob | 50 |
| directionality | bidi |
| hidden_act | gelu |
| hidden_dropout_prob | 0.1 |
| hidden_size | 512 |
| initializer_range | 0.02 |
| intermediate_size | 2048 |
| max_position_embeddings | 512 |
| num_attention_heads | 8 |
| num_hidden_layers | 4 |
| pooler_fc_size | 512 |
| pooler_num_attention_heads | 8 |
| pooler_num_fc_layers | 3 |
| pooler_size_per_head | 128 |
| pooler_type | first_token_transform |
| type_vocab_size | 2 |
| vocab_size | 27654 |
| n_context | 5 |
| context_rep_lr | 0.01 |
| pca_dim | 14 |
| contextual_warmup | 20,000 |

## D.2 BILINGUAL MODEL

As metioned in the paper, we use Wikipedia dump to pretrain the bilingual models. The Stanford CoreNLP tokenizer (Manning et al., 2014) is used to tokenize English, German, Spanish and Chinese data. And the spaCy tokenizer is used to tokenize Japanese data. All data are converted to lowercase. We convert Chinese data to simplified font to make it consistent with evaluation task datasets.

All the language models used in cross-lingual experiments were pretrained for 600,000 steps from scratch. Similar to our monolingual models, maximum number of senses ($S$ in word sense selection algorithm) was set to 1 for the first 20,000 steps, and the increased to 3 afterwards when pretraining Bi-SaELMo and Bi-SaELMo+Proj.[8] We use two NVIDIA V100 GPUs to pretrain each Bi-SaELMo model, which takes about 10 days to complete the training. See Table 8 for the hyperparameters used to pretrain Bi-SaELMo/Bi-SaELMo+Proj.

Table 8: Bilingual model hyperparameters: Bi-SaELMo/Bi-SaELMo+Proj

| Hyperparameter | Value |
|---|---|
| max_word_length | 50 |
| batch_size | 256 |
| n_gpus | 2 |
| bidirectional | True |
| char_cnn:embedding:dim | 16 |
| char_cnn:max_characters_per_token | 50 |
| char_cnn:n_characters | 261 |
| char_cnn:n_highway | 2 |
| dropout | 0.1 |
| lstm:cell_clip | 3 |
| lstm:dim | 4096 |
| lstm:n_layers | 2 |
| lstm:proj_clip | 3 |
| lstm:projection_dim | 512 |
| lstm:use_skip_connections | True |
| all_clip_norm_val | 10.0 |
| n_epochs | 6 |
| unroll_steps | 12 |
| n_negative_samples_batch | 8192 |
| n_context | 4 |
| cluster_proj_dim | 16 |
| pca_sample | 20,000 |
| remove_less_freqent_contexts | 0.1 |
| learning_rate | 0.2 |
| sense_learning_rate | 0.01 |

---

[8]Theoretically, in a reasonable range, it is expected that a larger $S$ would be more helpful to capture the fine-grained senses. However, due to limited computation power, we use only 3 here, and 5 for the monolingual models.

