# OpenReview forum: "Towards Multi-Sense Cross-Lingual Alignment of Contextual Embeddings"
_ICLR.cc/2021/Conference — Reject_

### Official Review · AnonReviewer3 · 2020-10-28
**Interesting Idea but Lacking Certain Experiment to Support the Claim**

**Rating:** 5
**Confidence:** 4

**Review:**


## Summary
Research Problem: ELMo or BERT does not model word-sense explicitly, and it introduces challenges in representation learning, especially in the multilingual setting.

This paper proposes a novel approach to model word-sense explicitly for (masked) language models (refer as LM for simplicity purpose) in an unsupervised fashion. It maintains a set of sense vectors for each word and the sense vectors are updated with online clustering approach. The correct sense is selected with the contextual representation and the model is trained to predict the correct sense. Finally, a sense-level translation loss is proposed for bilingual models. This paper presents both English monolingual model and bilingual model with 4 language pairs. It outperforms models without proposed component on word sense disambiguation and cross-lingual transfer on three tasks.


## Pros
1. The proposed sense-aware LM learns to model word-sense without any supervision in monolingual LM.
2. The proposed sense-aware LM and additional sense-level translation loss leads to improvement over regular bilingual models. In some cases, an extra projection step from previous work leads to further gain.

## Cons
1. This paper assumes the LM should be word-level, a relatively limited assumption when scaling to bigger corpus or more languages. It’s not straightforward to apply this approach to subword-level LM. It limits this paper to pretrain from scratch instead of fine-tune the existing BERT, multilingual BERT or XLM-R. It is not a problem for showcasing the proposed method in a controlled setting, but it limits the full potential of this paper, pushing state-of-the-art on top of the current best model. Additionally, this paper does not discuss the complexity of the proposed method.
2. While the experiment presents some evidence that the proposed method learns to model word-sense in an unsupervised fashion, it does not include any result comparison with prior work on incorporating word-sense into LM. As a result, it’s unclear how well the model learns word-sense compared to models with supervision.
3. It is unclear how much the sense-level translation loss contributes. This paper claims “... Bi-SaELMo is significantly better than Joint-ELMo, which shows that our sense-level translation pretraining objective improves cross-lingual embedding alignment”. However, the result presented in this paper does not support this claim. To support this claim, this paper should present **Bi-SaELMo without the sense-level translation loss**. Additionally, Bi-SaELMo assumes a bilingual dictionary compared to Joint-ELMo, yet the difference is not clearly discussed in the paper, especially in the result section.

## Reasons for score
Overall, I am leaning toward rejecting. While I find the sense-aware LM quite novel, the experiment presented does not support claim of contribution of sense-level translation loss. While the current result is promising, I find the limitation of the model not fully discussed and lack of comparison against other sense-aware LM in the monolingual setting. Hopefully the authors can clarify and address my concern in the rebuttal period.

## After revision
Thank you for answering my question!

> Our approach can be directly used to fine-tune pre-trained language models if word level tokenization is used.

It should be acknowledged that no multilingual word-based BERT exists as far as I know, partly due to the challenge with large vocabulary space and generalization. The sub-word approach you mentioned is a good proposal, but without evidence, I cannot assess whether it works. As a result, it's still a limitation of *this paper*, and should be clearly discussed in the paper.

> The complexity of the proposed method.

I am referring to the complexity w.r.t vocabulary size. The softmax normalization contains `nV` items, where `n` is the number of sense clusters and `V` is the vocabulary size. With the current limitation of word-based models, the vocabulary is already larger than sub-word models, the extra `n` factor cannot be disregarded.

> We train bilingual language models without sense-level translation loss (and without projection), denoted by Bi-SaELMo-NT, for ablation study.

Thank you for the ablation study! However, I cannot assess it as support for the claim as it's not an apple-to-apple comparison, and I cannot evaluate this paper based on future projection.

As a result, I have to maintain my rating based on the revision.

---

> ### Author Response · Authors · 2020-11-25
> **Response to Reviewer #3**
>
> ### This paper assumes the LM should be word-level, a relatively limited assumption when scaling to bigger corpus or more languages. It limits this paper to pretrain from scratch instead of fine-tune the existing BERT.
>
> Our approach can be directly used to fine-tune pre-trained language models if word level tokenization is used.  We also have plans to test methods to handle sub-word tokenization in our future work. For example, one possible way is to compute the average of sub-word level contextual representations as word level representation, which can then be used for sense prediction. So it can also be used to fine-tune existing BERT with this additional logic.
>
> &nbsp;
> ### The complexity of the proposed method.
>
> The computational complexity of our algorithm is linear with respect to the size of data, so our method is scalable to train on very large datasets.  We have added this to Section 3.1.
>
> &nbsp;
> ### It is unclear how much the sense-level translation loss contributes.
>
> We train bilingual language models without sense-level translation loss (and without projection),  denoted by Bi-SaELMo-NT, for ablation study. However, the models have only completed around ***85% of the total training steps*** (600,000 steps) by the last day of the rebuttal period. We follow the same setting as the zero-shot cross-lingual NER and sentiment classification experiments (Section 4.2) to evaluate its performance. The evaluation models are trained on English data and tested on German data. Below are the results for your reference:
>
> Zero-shot cross-lingual NER (de):
>
> |Model|F1|
> |---|---|
> |ELMo| 16.30|
> |Joint-ELMo| 56.49|
> |ELMo+Proj (Schuster et al., 2019)| 69.57|
> |Joint-ELMo+Proj (Wang et al., 2020)| 71.59|
> |Bi-SaELMo| 63.83|
> |Bi-SaELMo+Proj|72.19|
> |Bi-SaELMo-NT| 58.45|
>
> Zero-shot cross-lingual sentiment classification (de):
>
> |Model | books | music | dvd |
> |---|---|---|---|
> |ELMo|52.59 | 63.61 | 57.78 |
> |Joint-ELMo| 71.72 | 75.22 | 64.25 |
> |ELMo+Proj (Schuster et al., 2019)| 49.92 | 50.29 | 49.94 |
> |Joint-ELMo+Proj (Wang et al., 2020)| 75.74 | 72.25 | 72.25 |
> |Bi-SaELMo| 77.46 | 75.32 | 74.97 |
> |Bi-SaELMo+Proj| 70.84 | 66.25 | 68.99 |
> |Bi-SaELMo-NT| 72.41 | 74.95 | 63.60 |

---

### Official Review · AnonReviewer1 · 2020-10-28
**A good approach for sense-level alignment of contextual embeddings, but the experimental validation could be stronger**

**Rating:** 5
**Confidence:** 4

**Review:**

**Summary:**
This paper proposes the alignment of cross-lingual contextual embeddings not just at the word level, but at the sense level. It does this by relying purely on unaligned, unlabeled monolingual corpora used for pre-training, along with bilingual lexica. It does this by adapting the LM objective to be a sense-aware cross entropy loss, in which the sense is obtained by the use of a streaming k-means clustering algorithm combined with dimensionality reduction. If a bilingual lexicon is available, a sense-level translation objective can be added to encourage the model to predict the same sense in the other language (thereby encouraging identical senses of a word in the two languages to be closer together).

**Positives:**
* This is the first work (as far as I am aware) to perform sense-level alignment of contextual embeddings with only unlabeled, unaligned monolingual corpora and bilingual dictionaries.
* The proposed method works for both MLM and the next token prediction objectives. It should thus be applicable to most contextual embedding methods.
* The proposed method is extremely intuitive, but non-trivial. In particular, this work clearly presents its insights and well-justified tricks that seem to be crucial to the working of the proposed approach, for example: how simply selecting the vector $\\mathbf{w}\_{k',s}$ with the maximum dot product $\\mathbf{h}^T\_{k^*,s}\\mathbf{w}\_{k',s}$ does not work; how/why dimension reduction is crucial; the need to use $\\mathbf{h}\_{k,L}$ for sense selection even though $\\mathbf{h}\_{k-1,L}$ is used for next token prediction.

**Concerns:**
* In the case of the results presented for WSD, the ELMo baseline seems to be substantially worse than that reported in literature [1, 2] (ref: Table 2).
* While the paper talks about the applicability of the proposed method to both the next token prediction (in ELMo) and the MLM (in BERT) styles of language models, the experiments section seems very heavily focused on ELMo, not demonstrating the applicability of the method on BERT at all (aside from Table 2). Likewise, all the visualization in Appendix C focus solely on ELMo.

**Suggestions**
* In the section "Limitations of original training objectives", the authors state "We also observed the same with BERT embeddings.". This observation is central to the paper's argument for the need of the proposed approach in BERT, and elaborating on this with more analysis would have been nice.
* While the motivation behind the various tricks adopted were very well explained (refer: Positives), ablations/experimental numbers showing the failure of the proposed approach without these tricks would have helped strengthen this quantitatively.

**Questions:**
* Does the proposed method have any impact on the monolingual task (for example, for en on CoNLL?)
* The linear projection for cross-lingual alignment has been shown to not work on zero-shot sentiment classification. Do you have insight as to why this might be, why it adds noise to the embedding features?
* Can the proposed method directly be used to improve Bilingual Lexicon Induction? Perhaps by using monolingual data to train models with the sense-aware cross entropy loss and dynamic pruning to obtain the different sense embeddings, and then aligning these sense embeddings (Eg: with MUSE). Or if a bilingual lexicon is available, performing joint training followed by a supervised BLI approach like RCSLS [3]. Would this help learn a many-many mapping for BLI (where normally 1:many/many:many are difficult to achieve precisely because senses aren't taken into account)?
* Footnote 2 says that the dictionary can be learned in an unsupervised way (eg: using MUSE). Since the method here does not rely on a learned transformation, I'm assuming it was meant that we could use MUSE to generate the bilingual lexicon. How would this work with the proposed method, since unsupervised dictionaries generated with MUSE are usually 1:1/many:1 and tend to ignore multiple senses?

**Minor details:**
* Consider re-wording lines 7-10 in Algorithm 1 in terms of an argmax for conciseness, correctness and clarity

[1] Peters, Matthew E., et al. "Deep contextualized word representations." arXiv preprint arXiv:1802.05365 (2018).
[2] Hadiwinoto, Christian, Hwee Tou Ng, and Wee Chung Gan. "Improved word sense disambiguation using pre-trained contextualized word representations." arXiv preprint arXiv:1910.00194 (2019).
[3] Joulin, Armand, et al. "Loss in translation: Learning bilingual word mapping with a retrieval criterion." arXiv preprint arXiv:1804.07745 (2018).


======================================================================

**Update:**

I would like to thank the authors for their response. The lack of adverse impact of the proposed approach on monolingual tasks and the described ablations certainly help strengthen things on the experimental side.

However, my fundamental concerns still remain:
* I'm unsure why the ELMo baseline is so much worse than that reported in literature.
* With respect to the difficulty of training BERT, I can certainly empathize with the authors about the limited resources available in an academic setting. However, given that BERT was a key area of focus of the paper's methods section, showing the experimental results for BERT, even if BERT-tiny; note that BERT-tiny has just 2 layers and 128 hidden units, as opposed to 4 layers and 512 hidden units which this work uses (based on Section 4.1 and Table 7-- which corresponds to BERT-small, refer [here](https://github.com/google-research/bert#bert)), which should help reduce computational burden by quite a bit.

On account of this, I maintain my original rating of 5.

---

> ### Author Response · Authors · 2020-11-25
> **Response to Reviewer #1**
>
> ### Does the proposed method have any impact on the monolingual task (for example, for en on CoNLL?)
>
> We use LSTM-CRF to compare the performance of monolingual models on CoNLL 2003 English NER dataset. All models are trained and tested on English NER data. The average F1 of 5 runs are shown below. As we can see, the performance of these two models are very close on the monolingual NER task.
>
> |Model|F1|
> |---|---|
> |ELMo (baseline) |91.32|
> |SaELMo (ours) | 91.23|
>
> &nbsp;
> ### Can the proposed method directly be used to improve Bilingual Lexicon Induction?
>
> Yes, it is possible to improve Bilingual Lexicon Induction (BLI). We agree with the method  mentioned by the reviewer. Our model generates sense cluster centers (Algorithm 1) during language model pretraining. So it is possible to apply existing methods (e.g., MUSE/RCSLS) on sense cluster centers for BLI, and we will get sense level translations. Then the senses can be replaced with the corresponding words to generate the final result, which contains many to many mapping.
>
> &nbsp;
> ### Footnote 2 says that the dictionary can be learned in an unsupervised way (eg: using MUSE). Since the method here does not rely on a learned transformation, I'm assuming it was meant that we could use MUSE to generate the bilingual lexicon. How would this work with the proposed method, since unsupervised dictionaries generated with MUSE are usually 1:1/many:1 and tend to ignore multiple senses?
>
> Assume L1 and L2 are the two languages, it is possible to use MUSE to generate many:many dictionaries with the following steps:
> 1) Treat L1 as source language and L2 as target language, so we have many:1 mapping for multi-sense words in L2.
> 2) Treat L2 as source language and L1 as target language, so we have many:1 mapping for multi-sense words in L1;
> 3) Combine the dictionaries generated from the previous two steps.
>
> &nbsp;
> ### Very heavily focused on ELMo, not demonstrating the applicability of the method on BERT at all (aside from Table 2).
>
> Training a BERT model at a larger scale is too expensive for us (in academy), therefore we implemented our bilingual framework based on ELMo for evaluation. We have plans to extend the current approach to finetune BERT in the future work. We hope the reviewers understand our computation limitations and be inclusive as we believe big ideas are more important than big models.
>
> &nbsp;
> ### Ablations/experimental numbers showing the failure of the proposed approach without these tricks.
>
> Thanks for the suggestion. We ran a lot of experiments for ablation study when designing the algorithm. We will add more visualizations and ablation study results to the appendix. Below are some of our findings:
>
>
> 1. **Sense selection with/without the clustering algorithm (Algorithm 1):** Given any token to predict during language model pretraining, we directly compute the cosine similarity between the last layer contextual representation $h_{k^*,L}$ and the token’s sense vectors (i.e. context-dependent output embeddings) for sense selection. We count the number of times each sense vector is selected during model training, and find only one sense vector tends to be selected for each word, including the multi-sense words.
>
> 2. **With/without dimension reduction (Algorithm 2):** Since it is too time-consuming and inefficient to compare model performance for every change in the language model, we prepared a script to simulate the clustering algorithm. We sample sentences that contain “bank” from training data, and then generate the last layer contextual representations for “bank” with the checkpoint dumped in the middle of language models training (after 40,000 steps).  The contextual representations for “bank” are labeled with “bank_river” or “bank_other” by checking whether the sentences also contain “river”. We find most of the sentences that do not contain “river” correspond to the “financial institution” sense.
> We use the simulation script to compare the online clustering algorithm (for contextual representation clustering) with and without dimension reduction, and compute the percentage of “bank_river” and “bank_other” in each cluster. We find that without dimension reduction, the contextual representations of different senses tend to be classified to the same cluster. When dimension reduction is enabled, most of them can be correctly classified to different clusters.
>
> 3. **Compare different learning rates for clustering:** We use the same script as above to compare different learning rates. When we use the same learning rate as the neural model, we find that the cluster center vectors for each sense tend to change during the online clustering simulation, which means the original learning rate is too large. However, when we use a smaller learning rate (0.01 in our experiment), the cluster center vectors of different senses become stable. Therefore, it is necessary to set a separate learning rate for the clustering algorithm.

---

### Official Review · AnonReviewer4 · 2020-10-28
**Missing comparisons / distinctions with unsupervised sense models**

**Rating:** 4
**Confidence:** 3

**Review:**


Overview
===================

This is a really interesting paper, and it solves a big problem with cross-lingual representations: ignoring senses.

Pros: Interesting model, relatively unexplored problem, the quantitative evaluations in the paper are reasonable
Cons: Lack of comparisons with unsupervised sense algorithms, no qualitative evaluation of senses

Structure of algorithm
===================
The structure of the approach reminds me of the Yarowsky algorithm, has an EM-like flavor for building up the sense distributions.  It would be useful and interesting to discuss why it has this structure.  It would be useful to discuss a little more about why this was chosen rather than an end-to-end model that does the clustering and sense selection together.

Is this unsupervised?
===================

The paper argues that it is unsupervised, "Unlike these methods, our language models learn word senses in a fully self-supervised way".  However, I don't think this tells the whole story.  It needs a *multilingual dictionary* to know how many senses a word has and if it has a translation.

While MUSE (Conneau) provides unsupervised translations, the paper also adds  human-curated dictionary JMdict (including English, German, French, Russian and Dutch glosses), but the paper does not have a clear examination of the role of this additional supervision.

Given this level of supervision, it would be useful to have an explicit comparison with other supervised methods (e.g., have this model use WordNet or replace the other methods' dictionary with JMdict).

What are comparable models?
===================

There are, however, truly unsupervised sense induction models.  For instance, MUSE (Lee), GASI (Guo), and MSSG (Neelakantan).  Unfortuantely, this paper does not mention or compare with these models (they are, however, monolingual).

What are the relevant tasks?
===================

While WSD is a relevant task, it would be useful to have sense in context (Huang)  and/or interpretability (Guo) as tasks.  These tasks better capture whether unsupervised senses can determine a word meaning (without supervised data from SemCor) or make sense to an end user.

Either a formal or informal examination of what the sense clusters look like would help understand whether the algorithm is doing what it claims and to see if the output would be useful to humans (settings aside downstream tasks).

While the NER and XNLI tasks are reasonable, I'd rather see sense-specific evaluations.

Related Work not Cited
===================

Fenfei Guo, Jordan Boyd-Graber, Mohit Iyyer, and Leah Findlater. Which Evaluations Uncover Sense Representations that Actually Make Sense?. Linguistic Resources and Evaluation Conference, 2020.

Huang, E. H., Socher, R., Manning, C. D., and Ng, A. Y.  (2012). Improving word representations via global context and multiple word prototypes. In Proceedings of the Association for Computational Linguistics.

Lee, G.-H. and Chen, Y.-N. (2017). MUSE: Modularizing unsupervised sense embeddings. In Proceedings of Empirical Methods in Natural Language Processing.

Neelakantan, A., Shankar, J., Passos, A., and McCallum, A.  (2014). Efficient non-parametric estimation of multiple embeddings per word in vector space. In Proceedings of Empirical Methods in Natural Language Processing.

---

> ### Author Response · Authors · 2020-11-25
> **Response to Reviewer #4**
>
> ### It would be useful and interesting to discuss why it has this structure. It would be useful to discuss a little more about why this was chosen rather than an end-to-end model that does the clustering and sense selection together.
>
> In our approach, the language model and online sense clustering algorithm are trained in parallel. We have also tested the direct sense selection approach: given any token to predict during language model pretraining, directly compute the cosine similarity between the last layer contextual representation $h_{k^*,L}$ and the token’s sense vectors (i.e. context-dependent output embeddings) for sense selection. However, during experiments we found the model tends to select only one of the sense vectors. Since all of the sense vectors are used in softmax (Eq. 1 or Eq. 2) during each training step, their parameters will also be updated. Therefore, the sense vectors in softmax denominator will move in random directions, which is less controllable and will introduce more noise.
>
> Running an online clustering algorithm in parallel allows us to use the sense cluster centers (i.e. the weighted average of previous $h_{k^*,L}$ classified to each cluster) instead of sense vectors for sense selection, which is more robust to noise and helps the training process. The algorithm design rationales in section 3.1 discuss more advantages of this structure.
>
> &nbsp;
> ### Is this unsupervised?
>
> There may be some misunderstanding of our approach. Our approach for word sense modeling is fully unsupervised, where we use a clustering algorithm (Algorithm 1 on page 4) for sense clustering and sense selection. We do not use dictionaries here. Instead, the dictionaries are only used when computing the sense level translation loss (Section 3.2), which is for cross-lingual embedding alignment.
>
> &nbsp;
> ### Comparable models and related work.
>
> Thanks for suggesting the related work. We could not put enough discussion on some of these partially due to the constrained space.  Section 5 (related work) has been updated to add more discussion about these papers. For your convenience, we reiterate it here:
>
> Huang et al., (2012) learn word representations with both local and global context, and then apply a clustering algorithm to learn multi-prototype (multiple sense) vectors. Neelakantan et al., (2014) proposed an extension to the Skip-gram model that leverages a k-means clustering algorithm to learn multiple embeddings per word type. Suster et al. (2016) utilize both monolingual and bilingual information from parallel corpora to learn multi-sense word embeddings. Lee & Chen (2017) leverage reinforcement learning for modularized unsupervised sense level embedding learning. Guo et al. (2020) use Gumbel softmax for sense disambiguation when learning sense embeddings.
>
> &nbsp;
> ### Relevant tasks.
>
> Thanks for suggesting these relevant tasks for the monolingual model. Since our paper focuses more on cross-lingual embedding alignment, we only evaluated the monolingual model on the WSD task so as to leave more space for the 3 cross-lingual tasks. We will try to include more sense-specific evaluations. Besides, the visualisation of sense vectors in Appendix C also shows that our algorithm is doing what we have claimed.

---

### Official Review · AnonReviewer2 · 2020-10-28
**Good model design and good writing, however a bit lack motivation**

**Rating:** 6
**Confidence:** 5

**Review:**

This paper proposes to introduce multiple senses into pre-trained models. The proposed method selects senses dynamically while pretraining the model and applies a sense-aware cross-entropy loss for pretraining. This paper further proposes to jointly pre-train a sense-aware cross-lingual model with sense-level translation. The proposed model yields better performance than the baseline models under both monolingual and cross-lingual setting.

Strength:

1) The pipeline is designed very well and covers many aspects: taking care of the pruning while training; introduce projections to reduce parameters; from monolingual to cross-lingual setting.
2) The paper is well written and easy to follow.
3) Evaluation on three downstream tasks show significant improvements over baseline models, and for some case  (NER de), it performs even better than larger cross-lingual models.

Weakness:

1) Should cite more multi-sense papers (at least more papers before 2018).

Especially:

https://www.aclweb.org/anthology/P12-1092.pdf
https://www.aclweb.org/anthology/D14-1113/
(both uses clustering methods to select senses)

https://www.aclweb.org/anthology/N16-1160.pdf
https://www.aclweb.org/anthology/D17-1034/
https://www.aclweb.org/anthology/2020.lrec-1.214.pdf
(all use similar softmax form to predict senses as the prediction task in your sense-aware cross-entropy loss)

2) The proposed model outperforms the baseline models significantly.  However, the baseline models are pretty out-dated. And the scale is quite small. It is true that your model performs better on NER for DE than larger cross-lingual models. However, in most cases, it performs much worse than large-scale pre-trained cross-lingual models (https://arxiv.org/pdf/1901.07291.pdf). Have you tried to apply your methods to large-scale models? Is it possible that large-scale models have already captured the sense information within the context and the context provides enough information for disambiguations? Therefore introducing senses won't bring more capacity to the model?

---

> ### Author Response · Authors · 2020-11-25
> **Response to Reviewer #2**
>
> ### Should cite more multi-sense papers.
>
> Thanks for suggesting the related work. We have updated Section 5 (related work) to add more discussion about these papers. For your convenience, we reiterate it here:
>
> Huang et al., (2012) learn word representations with both local and global context, and then apply a clustering algorithm to learn multi-prototype (multiple sense) vectors. Neelakantan et al., (2014) proposed an extension to the Skip-gram model that leverages a k-means clustering algorithm to learn multiple embeddings per word type. Suster et al. (2016) utilize both monolingual and bilingual information from parallel corpora to learn multi-sense word embeddings. Lee & Chen (2017) leverage reinforcement learning for modularized unsupervised sense level embedding learning. Guo et al. (2020) use Gumbel softmax for sense disambiguation when learning sense embeddings.
>
> &nbsp;
>
> ### The proposed model outperforms the baseline models significantly. However, the baseline models are pretty out-dated. And the scale is quite small.
>
> Training a BERT model at a larger scale is too expensive for us (in academy), therefore we implemented our bilingual framework based on ELMo for evaluation. We have plans to extend the current approach to finetune BERT in the future work. We hope the reviewers understand our computation limitations and be inclusive as we believe big ideas are more important than big models.

---

### Decision · Program_Chairs · 2021-01-07
**Final Decision**

**Decision:**

Reject

**Comment:**

This paper investigates how to align word senses across languages. This has not been studied much as past work has primarily considered aligning word (embeddings) across languages. The paper is well written and well motivated. Unfortunately the empirical results are not very strong. The baselines are somewhat low and the gains are modest (the excuse that it is difficult to train BERT-sized models in academia is acknowledged). Overall, there is not enough support for acceptance at such a competitive venue as ICLR.